# Vaccine Hesitancy among Master’s Degree Students in Nursing and Midwifery: Attitude and Knowledge about Influenza Vaccination

**DOI:** 10.3390/ijerph17197191

**Published:** 2020-10-01

**Authors:** Claudia Mellucci, Andrea Tamburrano, Fabiana Cassano, Caterina Galletti, Anna Sguera, Gianfranco Damiani, Patrizia Laurenti

**Affiliations:** 1Section of Hygiene, Woman and Child Health and Public Health, Università Cattolica del Sacro Cuore, 00168 Rome, Italy; claudia.mellucci@gmail.com (C.M.); fabiana.cassano94@gmail.com (F.C.); caterina.galletti@unicatt.it (C.G.); gianfranco.damiani@unicatt.it (G.D.); patrizia.laurenti@unicatt.it (P.L.); 2Woman and Child Health and Public Health, Fondazione Policlinico Universitario A. Gemelli IRCCS, 00168 Rome, Italy; anna.sguera@policlinicogemelli.it

**Keywords:** nurses, midwives, students, influenza vaccination, attitude, survey, public health, preventive medicine

## Abstract

Influenza vaccination among healthcare workers may reduce morbidity and protect fragile patients. Most of the evidence concerning the vaccine hesitancy of healthcare workers reported lack of knowledge and wrong attitude. The aims of this study were to explore the knowledge and attitudes about influenza vaccination among master’s degree students in Nursing and Midwifery, and to evaluate the effectiveness of their involvement in the hospital vaccination campaign in order to increase intention to receive immunization. The students of nurses and midwives were involved in the vaccination sessions of the 2018–19 hospital campaign. They were recruited to complete an online survey. Students of the 2nd year (involved in the vaccination campaign) and the 1st year (not involved) were compared. Descriptive and inferential statistics were performed for data analysis. Students who intend to receive influenza vaccination in the following year registered a percentage of 83.6% and showed an overall attitude of 66.8%. The involvement of the students in the vaccination campaign led to a significant increase in their positive vaccination attitude (80.9% vs. 87.0%) and in their intention to receive flu vaccination in the following year (67.7% vs. 100%). A positive attitude towards vaccinations was observed by nurses and midwives. Their involvement in the planning and activities during the vaccination campaign could positively influence their opinions and intention to receive vaccination.

## 1. Introduction

Healthcare professionals, involved in patient care and management, have a greater risk of acquiring infection than the general population; moreover, being constantly in contact with several people (patients, family members and other healthcare workers) also makes them real carriers of the infection [1]. According to the literature, most of the recommendations strongly aim to promote the influenza vaccination of all healthcare workers (HCWs), with special consideration to those who provide assistance in departments with a higher risk of acquiring and transmitting the infection [2,3].

Despite the above assumptions, worldwide vaccination coverage of HCWs is lower than the standards set by the World Health Organization [4]; among the general population, the minimum objective for flu vaccination coverage is 75% and the optimal target is 95% [5].

In Italy, HCW vaccination coverage of the last season (2018–19) was the same as the previous years—15.6%; in Europe, Belgium and the United Kingdom were closer to recommended coverage rates, at 60% and 65% respectively. In the United States of America, vaccination coverage among HCW reached 78.4% during the 2017–18 season; for the last season (2018–19), vaccination coverage among member states rose to 91% [6], and 90.5% of USA nurses received vaccination. Nurses outside of the USA displayed less interest in influenza vaccine during flu seasons [7,8].

Furthermore, there is an increasing body of scientific evidence concerning the vaccine hesitancy of the nurses, most of them reporting motivations linked to the origin country, religion, knowledge, and contextual influence [9].

Despite training programs carried out among nurses and other healthcare personnel, poor attitude regarding vaccination is found [10]. These data also impact the general population, recording significant associations between vaccination coverage of the HCWs and cases of infection in hospitalized patients [11].

From the literature, personal awareness and knowledge of the HCWs are considered the main factor for increasing vaccinations and spreading recommendations to the patients [12], as well as the self-protection of HCWs and the protection of family members or colleagues [13]. On the contrary, the lack of concerns and perception of risk about the flu illness are the main reasons for the reluctance to influenza vaccine [14].

Since the flu season of 2015–2016, our high-care complexity university hospital has organized an influenza vaccination campaign aimed at increasing the vaccination rate. For the 2018–2019 campaign, some improvements were implemented: on-site vaccination at all the hospital wards, non-economic incentives for the vaccinated personnel (1 h off work), and the active participation of nurses and midwives as staff vaccinators (who attended influenza vaccination training).

The purpose of the study is to explore the knowledge, opinions and attitudes on flu vaccination of the master’s degree students in Nursing and Midwifery, and to evaluate the effectiveness of the students’ involvement from the second year course in the 2018–2019 hospital campaign in order to improve their attitude toward vaccination and their intention to receive vaccination in the following year.

## 2. Materials and Methods

### 2.1. Nurses and Midwives in the Hospital Vaccination Team

For the last vaccination campaign of Gemelli University Hospital of Rome, a multidisciplinary vaccination team was established; it was made up of physicians, nurses, and midwives of the second year of the master’s degree in Nursing and Midwifery (a two-year course for professional nurses and midwives). A mentor, identified in the group, had the assignment to follow and motivate vaccination-oriented colleagues. Training programs regarding the importance of vaccination were also implemented during the master’s degree session among the second-year students. They consist of 20 h training with the aim of both improving knowledge and acquiring practical skills regarding vaccines against the flu. The students were involved in the vaccination sessions and were asked to provide written documentation of their activities (attitudes, number of vaccinations performed, and level of satisfaction regarding the operation of the vaccination team). Vaccination activities during the hospital campaign were included in the practical training as a part of the master’s degree program, and a final certification was provided.

### 2.2. Design and Data Collection

To evaluate the effectiveness of the new measures on the attitude of nurses and midwives regarding vaccination and their intention to receive vaccination, a cross-sectional study was carried out at the end of the 2018–2019 vaccination campaign through the submission of a validated questionnaire. In order to assess the effectiveness of the newly implemented measures on the students’ attitude toward vaccination, the results were evaluated by comparing the answers of the second-year students (who participated in the last vaccination campaign) with those of the first year (who did not participate).

Data were collected between April and June 2019. The link to the survey was sent to the e-mail addresses of the 1st and 2nd year master’s degree students, and it was completed anonymously in electronic format. All the answers were finally collected in an “.xlsx” database.

### 2.3. Survey Tool

The validated tool [15] is an anonymous questionnaire, entirely composed of closed-ended questions (Appendix A). The questionnaire consists of 3 sections:

Section A investigates the opinions and knowledge regarding influenza and flu vaccination. This section provides answers on a 4-point Likert scale. Answers “totally disagree” and “partially disagree” were considered “disagreement”, answers “totally agree” and “partially agree” were considered “agreement”.

Section B investigates the adherence to the flu vaccination and the preferences of the vaccine administration. This section provides multiple or dichotomous answers (yes/no).

Section C collects the responders’ socio-demographic characteristics, their academic titles, and their opinion on the 2018–19 hospital vaccination campaign. This section provides multiple and dichotomous (yes/no) answers and two 10-point numerical scale questions.

### 2.4. Data Analysis

A descriptive analysis of the questions in sections A, B and C was made, reporting frequencies, percentages, mean, median and standard deviation. In addition, some questions from section A (1–5, 8, 14, 15) were filed together to analyze the general attitude of nurses and midwives regarding flu vaccination. The results were stratified by gender, academic title, and year of study.

Furthermore, answers to the question 1 and 2 in section B were compared to the answers to the question 8 and 9 in section C; the cited questions were also compared with the values obtained from the general vaccination attitude analysis.

The relationship between the above items was tested by Student’s t-test for the quantitative variables, by chi-square test, Fisher’s exact test and Yates’s chi-square test for categorial ones. Finally, the relationship between the students’ opinions on the hospital vaccination campaign (question 8 and 9—section C) and other variables was tested by the Mann–Whitney U test.

The significance level was set at 0.05. Statistical analysis was conducted with STATA software ver.13.1 (Statacorp, College Station, TX, USA).

### 2.5. Ethical Considerations

All subjects gave their informed consent agreement for participation. The study is compliant with the Hospital Ethical Committee Standards (Prot. 41409/18, ID: 2263), in accordance with the Helsinki Declaration and EU Regulation 2016/679 (GDPR) concerning the processing of personal data.

## 3. Results

### 3.1. Sample Characteristics

The sample is composed of master’s degree students in Nursing and Midwifery. The students who completed the questionnaire are 61 of 66 (response rate: 92.4%). There were 31 first year students (50.8%) and 30 s year students (49.2%). Further characteristics of the sample are shown in Table 1.

### 3.2. Knowledge on Vaccination

Regarding the students’ knowledge and awareness on vaccination, 59 (96.7%) of them were aware that vaccination is recommended for HCWs, 57 (93.4%) reported that flu is a risky disease, while 56 (91.8%) were aware of the possibility to transmit the flu to their patients.

On the other hand, 24 of them (39.3%) reported that the vaccine could cause influenza and 14 (23.0%) that the adjuvant has serious side effects, while 7 (11.5%) reported that the vaccine has serious side effects. There were 31 (50.8%) students who reported that their colleagues do not receive vaccination, while 5 (8.2%) were against vaccination. Lastly, six responders (9.8%) were afraid of needles. A complete panel of the answers is illustrated in Table 2.

### 3.3. Attitude toward Vaccination

An overall assessment considering interviewees’ attitudes towards vaccination was made. The results show an agreement rate of 66.8% (Figure 1).

Differences related to gender, academic titles and year of study are shown in Table 3.

To assess the effectiveness of the new intervention campaign on the Nursing and Midwifery students, vaccination attitude between the 2nd year (involved in the vaccination campaign) and the 1st year students (not involved) was compared, recording a statistically significant difference (*p* < 0.01). The results are shown in Figure 2.

### 3.4. Vaccination Status

Considering the flu vaccination status of the two groups of students together, 19 responders (31.1%) declared to have been vaccinated three years ago, 27 (44.3%) two years ago, and 35 (57.4%) in the last flu season. Fifty-one responders (83.6%) intended to receive the seasonal influenza vaccination in the following year, while 10 (16.4%) gave a negative response.

Analyzing the reasons given by those who were not vaccinated the last flu season, 10 students (16.4%) believed that it is better to get the flu rather than get vaccinated, and 5 (8.2%) were completely against vaccination.

On the other hand, 57 responders (93.4%) perceived the importance of vaccine protection for themselves, and 59 (96.7%) were aware of the protection transmitted to their families and acquaintances. Further questions from section B are shown in Table 4.

### 3.5. Opinions on the Hospital Vaccination Campaign

Analyzing the answers to question 8 of section C “How useful is it to implement a campaign aimed at making the hospital’s health personnel aware of the flu vaccination?”, a median rating of 10 (from 3 to 10) points was reported.

Moreover, those who showed a vaccination attitude in the previous answers expressed a median of 10 (from 3 to 10), and those who had no attitude showed a median of 10 (from 3 to 10).

Likewise, those who were vaccinated last year expressed a median of 10 (from 6 to 10), and those who were not vaccinated showed a median of 10 (from 3 to 10).

Considering the answers to the question 9 of section C “How effective was the hospital’s last influenza vaccination campaign?” a median rating of 8 (from 2 to 10) points was reported.

Those who showed a vaccination attitude expressed a median of 8 (from 2 to 10), while those who had no attitude showed a median of 7 (from 2 to 10). This difference (1 point) is not statistically significant (*p* = 0.06).

Finally, those who were vaccinated last year expressed a median of 8 (from 5 to 10), while those who were not vaccinated showed a median of 7 (from 2 to 10). This difference (1 point) is statistically significant (*p* < 0.01).

### 3.6. Effectiveness of the Intervention

The university hospital’s vaccination campaign involved students of the second year of the master’s degree course in Nursing and Midwifery, who supported the dedicated staff for HCWs’ vaccination.

In term of vaccination rate, our study shows statistically significant differences between first- and second-year students (*p* < 0.001; OR = 10.02). The first-year group was vaccinated the previous year in 10 cases (32.3%), while second-year group (involved in the campaign) was vaccinated in 25 cases (83.3%).

Considering the intention to receive influenza vaccination in the following year, the second-year students agreed in 30 cases (100%), while the students of the first year agreed in 21 cases (67.7%). The difference is statistically significant (*p* < 0.001; OR = 21.18).

Analyzing the general vaccination attitude, the students involved in the last campaign showed a positive attitude in 631 answers (87.0%) vs. 573 answers (80.9%) of those not involved. The difference is statistically significant (*p* < 0.01; OR = 1.58).

Moreover, the involved students evaluated the usefulness of “implementing a campaign aimed at making the hospital’s health personnel aware of the flu vaccination” (question 8, section C) with a median score of 10 (from 6 to 10), and those not involved presented a median score of 10 (from 3 to 10).

Finally, the involved students evaluated the effectiveness of the “hospital’s last influenza vaccination campaign” (question 9, section C) with a median score of 8 (from 6 to 10) vs. a median score of 7 (from 2 to 10) given by the students not involved. The difference (1 point) is statistically significant (*p* = 0.001).

## 4. Discussion

Through this study, a positive vaccination attitude was observed among nurses and midwives. Most of them (83.6%) intended to receive influenza vaccination in the following year and showed an overall positive vaccination attitude of 66.8%.

Not being in a risk category is the main reason for not being vaccinated among the students at our university hospital, who showed an accordance of 16.4%; a higher percentage (39.9%) was observed in a study conducted among the students of the Careggi University of Florence [16]. Furthermore, a general lack of knowledge about influenza and vaccines was found; recent literature demonstrate that training programs could be useful to spread knowledge and correct information aimed to improve vaccination adherence [17,18,19,20].

The main positive statement given by students who showed a good attitude to vaccination was the protection offered by the vaccine (“Not to get the flu” and “To protect my cohabitants/contacts”, 95.0%). These results are strongly confirmed by various publications, where high percentages of agreement relating to personal and family member protection are reported (90% [14]; 77% [21]; 86% [22]).

Regarding the opinions on the usefulness of a vaccination campaign aimed at promoting the vaccine for health workers (question 8), we found no significant differences between the first year and second year students. Instead, the opinions on the effectiveness of the hospital’s last flu vaccination campaign (question 9) showed significant differences between the two groups in terms of vaccination status and involvement in the campaign. Moreover, the analysis showed a higher approval of the vaccination campaign, promoted by the hospital, in 2nd year students (+1 point; *p* = 0.001).

Finally, the involvement of the students in the vaccination campaign has led to a moderate but significant increase in their vaccination attitude (80.9% vs. 87.0%; *p* < 0.01), a greater and significant increase in their vaccination rate of from the previous year (32.3% vs. 83.3%; *p* < 0.001), and an increase in their intention to receive influenza vaccination in the following year (67.7% vs. 100%; *p* < 0.001).

These results show how the intervention on the students could have a positive influence on their attitude and beliefs regarding influenza vaccination.

These findings are confirmed in several studies [23,24] in which the involvement of nursing staff in training and vaccination campaigns for HCWs gives positive results in terms of adherence. The hospital staff involvement in the vaccination campaign should be implemented in hospitals to increase vaccination adherence among health personnel.

### Limitations

This study has some limitations. A first limitation is the small sample size, which did not allow for more in-depth assessments of vaccination skills and attitudes of the nurses and midwives, although the questionnaire used allowed a comprehensive general screening.

Moreover, five students did not complete the questionnaire: by excluding them from the data, it was not possible to assess their positive or negative opinion on vaccination.

Another limitation is represented by the setting: this monocentric study showed data that referred to a single university hospital. Further studies are recommended to explore other academic realities.

Furthermore, the responders are mainly young healthcare workers, with relatively poor experience; the results obtained may not be representative of the entire nursing and midwifery categories.

A last limitation is related to the students’ involvement in the campaign: the comparisons were possible only between students of two different years of course. Among the second-year students, compulsory vaccination training did not allow for the selection of a control group.

## 5. Conclusions

The questionnaire proved to be an effective tool to detect HCWs’ knowledge, practices, opinions and preferences on influenza and flu vaccination, and to collect useful information and suggestions to guide future strategies for promoting influenza vaccination among HCWs.

Relating to the poor knowledge found among nurses and midwives, as already demonstrated in a study conducted in an Italian university hospital [25], multidisciplinary training programs or post-graduate courses could improve the attitude of future health professionals towards flu vaccination [26].

This study shows how the involvement of nursing and midwifery students in the planning and execution of the vaccination campaign can influence their opinions and beliefs on flu vaccination among HCWs. Moreover, the vaccination training is correlated with an increase in their vaccination attitude, increasing from 67.7% in the 1st year students to 100% in the 2nd year (*p* < 0.01). Differently, the impact on students’ intention to receive vaccination during the following campaign and the future ones was moderate, increasing from 80.9% to 87% (*p* < 0.001).

A well-planned vaccination campaign with a multidisciplinary approach could serve to improve the HCWs’ attitudes, especially when involved in the campaign implementation. The real intention to be vaccinated, instead, is a difficult goal to achieve only through a vaccination campaign. As confirmed by the latest literature, different strategies could be used to improve vaccination attitudes, such as the mandatory vaccination policies [27] or some economic and non-economic incentives for the vaccinated personnel [28].

## Figures and Tables

**Figure 1 ijerph-17-07191-f001:**
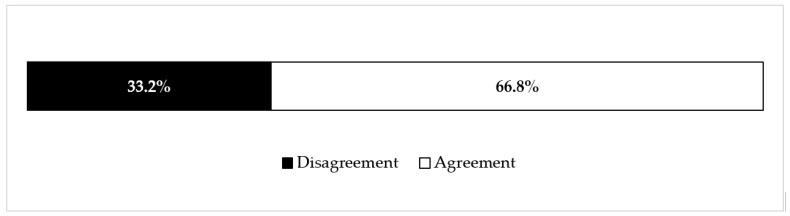
Responders’ vaccination attitude (%).

**Figure 2 ijerph-17-07191-f002:**
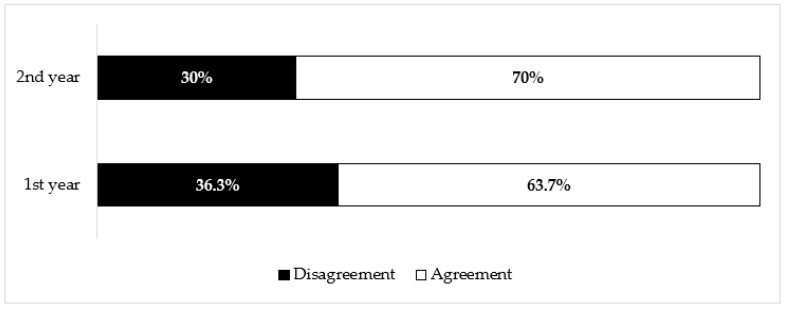
Responders’ vaccination attitude (%) stratified by year of study.

**Table 1 ijerph-17-07191-t001:** Sample characteristics (number of participants, mean age, and *p*-value) stratified by gender, academic title, living conditions and year of study.

Variables	N (%)	Mean Age ± SD	*p*-Value
Gender			
F	45 (73.8%)	32.5 ± 11.4	0.470
M	16 (26.2%)	30.3 ± 6.6	
Academic title			
Degree	41 (67.2%)	30.5 ± 11.0	0.098
Post-graduate courses	20 (32.8%)	34.8 ± 8.4	
Living conditions			
Living with children under 9 years old	11 (18.0%)	36.6 ± 7.0	0.546
Living with people over 65 years	19 (31.1%)	38.6 ± 12.9	
Living with people with chronic disease	13 (21.3%)	41.9 ± 13.6	
Year of study			
1st	31 (50.8%)	33.7 ± 12.2	0.163
2nd	30 (49.2%)	30.0 ± 7.7	

**Table 2 ijerph-17-07191-t002:** Sample’s answers to Section A of the questionnaire (number and percentage).

Questions	N (%)			
	Disagreement		Agreement	
	Totally Disagree	Partially Disagree	Partially Agree	Totally Agree
1. Influenza is a risky disease	0 (0.0%)	4 (6.6%)	26 (42.6%)	31 (50.8%)
2. It is better to get sick than to get vaccinated	43 (70.5%)	8 (13.1%)	10 (16.4%)	0 (0.0%)
3. The flu vaccine has serious side effects	35 (57.4%)	19 (31.1%)	7 (11.5%)	0 (0.0%)
4. The flu vaccine can cause influenza	26 (42.6%)	11 (18.0%)	18 (29.5%)	6 (9.8%)
5. The flu vaccine is effective	1 (1.6%)	1 (1.6%)	26 (42.6%)	33 (54.1%)
6. The adjuvant increases the effectiveness of the vaccine	5 (8.2%)	10 (16.4%)	30 (49.2%)	16 (26.2%)
7. The adjuvant has no serious side effects	3 (4.9%)	15 (24.6%)	28 (45.9%)	15 (24.6%)
8. I am against vaccination	54 (88.5%)	2 (3.3%)	2 (3.3%)	3 (4.9%)
9. My colleagues do not get vaccinated	18 (29.5%)	12 (19.7%)	22 (36.1%)	9 (14.8%)
10. Healthcare professionals must get vaccinated	0 (0.0%)	2 (3.3%)	7 (11.5%)	52 (85.2%)
11. I am afraid of needles	51 (83.6%)	4 (6.6%)	3 (4.9%)	3 (4.9%)
12. I do not get vaccinated, so if I get sick, I can stay at home	57 (93.4%)	2 (3.3%)	2 (3.3%)	0 (0.0%)
13. It is likely that I transmit the flu	3 (4.9%)	2 (3.3%)	12 (19.7%)	44 (72.1%)
14. By getting vaccinated, I protect myself from the flu	2 (3.3%)	2 (3.3%)	17 (27.9%)	40 (65.6%)
15. By getting vaccinated, I protect my cohabitants/contacts from the flu	1 (1.6%)	1 (1.6%)	13 (21.3%)	46 (75.4%)
16. My cohabitants/contacts expect me to be vaccinated against the flu	3 (4.9%)	12 (19.7%)	21 (34.4%)	25 (41.0%)
17. The adjuvant has serious side effects	29 (47.5%)	18 (29.5%)	13 (21.3%)	1 (1.6%)
18. I know where to get the flu vaccination	0 (0.0%)	2 (3.3%)	9 (14.8%)	50 (82.0%)
19. My district promotes flu vaccination	1 (1.6%)	5 (8.2%)	13 (21.3%)	42 (68.9%)

**Table 3 ijerph-17-07191-t003:** Vaccination attitude stratified by gender, academic title, and year of study.

Variables	N (%) of Answers					
	Disagreement		Agreement			
	Totally Disagree	Partially Disagree	Partially Agree	Totally Agree	Tot.	*p*-Value
Gender						
F	68 (18.9%)	53 (14.7%)	73 (20.3%)	166 (46.1%)	360	0.738
M	27 (21.1%)	14 (10.9%)	27 (21.1%)	60 (46.9%)	128	
Academic title						
Degree	62 (18.9%)	45 (13.7%)	63 (19.2%)	158 (48.2%)	328	0.633
Post-graduate courses	33 (20.6%)	22 (13.8%)	37 (23.1%)	68 (42.5%)	160	
Year of study						
1st	45 (18.1%)	45 (18.1%)	59 (23.8%)	99 (39.3%)	248	0.002
2nd	50 (20.8%)	22 (9.2%)	41 (17.1%)	127 (52.9%)	240	

**Table 4 ijerph-17-07191-t004:** Sample’s answers to questions 3–9 from section B of the questionnaire.

Questions from Section B	N (%)
N 3: If you were not vaccinated against seasonal flu last year, what was the reason(s)?	
I am not in a risk category	10 (16.4%)
I was worried about side effects	3 (4.9%)
The vaccine is not effective	2 (3.3%)
The vaccine causes the flu	1 (1.6%)
I never get the flu	8 (13.1%)
The place/time of the vaccination was not suited to my schedule	3 (4.9%)
I had no time	5 (8.2%)
I forgot to get vaccinated	5 (8.2%)
I am afraid of needles	1 (1.6%)
No one informed me about vaccination	4 (6.6%)
I have never been vaccinated before	10 (16.4%)
Other: suffering from diseases incompatible with vaccination	3 (4.9%)
N 4: If you were vaccinated against seasonal flu last year, what was the reason(s)?	
Not to get the flu	31 (50.8%)
To protect my cohabitants/contacts	35 (57.4%)
I get vaccinated every year	14 (23.0%)
I have been ill with influenza in the past	16 (26.2%)
The place/time for vaccination was appropriate to my schedule	23 (37.7%)
They advised me to do it	18 (29.5%)
I felt I had to do it	31 (50.8%)
N 5: If you were vaccinated last year, who provided the vaccination?	
The doctor of the hospital vaccination service	15 (24.6%)
The doctor of the district vaccination service	5 (8.2%)
The general practitioner	6 (9.8%)
A colleague	9 (14.8%)
I was not vaccinated last year	26 (42.2%)
N 6: If you intend to receive influenza vaccination, who would you like to receive the vaccination from?	
The doctor of the hospital vaccination service	20 (32.8%)
The doctor of the district vaccination service	7 (11.5%)
The general practitioner	11 (18.0%)
A colleague	21 (34.4%)
I do not want to get vaccinated	2 (3.3%)
N 7: The vaccination is to be recommended:	
to people over 65 years	57 (93.4%)
to pregnant women after the first trimester	38 (62.3%)
to health professionals	58 (95.1%)
to oncology patients	43 (70.5%)
to patients with diabetes and cardiac conditions	49 (80.3%)
to patients with COPD * and renal failure	49 (80.3%)
to children and healthy young people	35 (57.4%)
N 8: What kind of vaccine do you prefer?	
Adjuvanted	27 (44.3%)
Not adjuvanted	4 (6.6%)
I do not know/they are the same	27 (44.3%)
I do not want to get vaccinated	3 (4.9%)
N 9: What kind of vaccine administration would you like to receive?)	
Intradermal	14 (23.0%)
Intramuscular/subcutaneous	35 (57.4%)
I do not know/they are the same	11 (18.0%)
I do not want to get vaccinated	1 (1.6%)

* Chronic obstructive pulmonary disease.

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
