# Peer review of "Vaccine Hesitancy among Master’s Degree Students in Nursing and Midwifery: Attitude and Knowledge about Influenza Vaccination"

_ijerph, 2020, doi:10.3390/ijerph17197191_

Round 1
Reviewer 1 Report
The paper by Mellucci, et al analyzed the role of participation and active involvement in a hospital vaccination campaign on increasing influenza vaccine uptake and attitudes. Second year students in the nursing and midwifery master’s program had the opportunity to be directly involved with the vaccination campaign though planning and activities. While the authors couldn’t specifically tease apart survey takers who participated in the campaign, and those who didn’t, they did compare first year students in a master’s program (who would not have had the opportunity to participate) and second year master’s students (who would have had the opportunity to participate). Despite this, they saw a modest but significant impact on both the intent to receive the vaccine and a positive attitude towards vaccination.
Major Revisions:
- It is hard to interpret the data without knowing more about the hospital vaccination campaign. A brief summary of the campaign efforts should be included in the material and methods, introduction, or discussion section.
Minor Revisions:
- The term attitude is used frequently to describe a positive attitude about vaccination, the authors should be more precise with how they use the word attitude, because it can be both positive or negative depending on the context. For example in the abstract, line 21, “an increase in attitude” should probably be changed to “an increase in positive attitude” (or something similar).
- The survey has multiple options for answering “Other” in part B. Did any students select this option? If so, how many, and were the results relevant to this study?
- The authors should address the fact that the 5 students who did not complete the questionnaire are potentially biased against vaccination. By excluding them and not accounting for them in the data, they results could be skewed toward the students that already viewed vaccinations positively or were at least open to viewing them more positively. This should at least be addressed in the Limitations section of the manuscript.
Author Response
Major Revisions:
- It is hard to interpret the data without knowing more about the hospital vaccination campaign. A brief summary of the campaign efforts should be included in the material and methods, introduction, or discussion section.
We explain the efforts of the hospital vaccination campaign at the end of the Introduction section (row 58-62). Following your suggestion, we also added further clarification about the involvement of nurses and midwives in the campaign in the first paragraph of Materials and Methods section (row 70-80).
Minor Revisions:
- For example in the abstract, line 21, “an increase in attitude” should probably be changed to “an increase in positive attitude” (or something similar).
We adjusted the sentence in the abstract.
- The survey has multiple options for answering “Other” in part B. Did any students select this option? If so, how many, and were the results relevant to this study?
We did not receive any “Other” answer from the students.
- The authors should address the fact that the 5 students who did not complete the questionnaire are potentially biased against vaccination.
We added an analysis of the limit you correct report in the limitations section (row 240-241).
Reviewer 2 Report
Review of Ijerph-931928
Specific issues:
Page 1, line 16. May want to consider changing ‘in increasing intention’ to ‘to increase intention …..
Page 1, line 18. Please consider making ‘also including students of the previous year (not involved) to its own sentence and explain in succinctly such as ‘Students from the previous year also completed an online survey and used as a control group’
Page 1, line 19. Numerals at the beginning of a sentence must be spelled out.
Page 1, line 21 – the rest of the abstract is written in past tense, please change ‘has led’ to ‘led’.
Page 1, line 24 and 25. Please change to past tense.
Page 1, line 33. Change to according ‘to’ literature….
Page 1, line 36. ‘despite the premises’ This does not make sense. Consider changing ‘premises’ to another word that more adequately describes what you are saying.
Page 1, line 39. Are you saying that in Italy, HCW vaccination coverage is 15.6%? rewrite for clarity.
Page 1, line 40. Delete ‘the countries’.
Page 1, line 41. Delete ‘on the other hand’. Not needed.
Page 2, lines 43-44. ‘While ….in the USA’ is not a complete sentence. Rewrite.
Page 2, line 45. This whole sentence needs to be rewritten for clarity. Such as ‘nurses outside of the USA have lower attitudes to influenza vaccine during flu seasons’.
Page 2, lines 43-48. Combine this information into one paragraph.
Page 2, line 52. Delete ‘on’.
Page 2, lines 53-54. What is ‘knowledge of the operators’? who are the operators? Also delete ‘as’ after considered.
After the above Page 2, lines 53-54 and through the rest of this document, all the yellow highlights in the manuscript are syntax errors and need to be rewritten for clarification or for grammar editing.
My questions will be in Green highlights, that are not necessarily considered syntax errors.
Page 3. Line 83. Would the survey be completed rather than submission?
Page 3, lines 107-110. This information on the difference between students in first year vs second year needs to be stated in section 2.3 Design and data collection.
Page 3, line 124. This says the course. Is it one course or the program or year of study?
Page 4, Table 1. Is there a difference between Degree and Postgraduation titles? Please explain in the narrative text.
Page 5, line 141. Please tell us which questions were evaluated in this assessment. One question? Several, please be specific.
Page 5, Table 2. Does this reflect the answers of all the students? Would it make sense to have two tables, one for the 1st year students and one for the 2nd year students?
Page 6, lines 147-150. I would like to see the questions that ‘vaccination attitude’ is derived from. Are they the numbers and percentages in Table 3 and/or Figure 2? Please tell us more as to how you ascertained this information.
Page 6, lines 154-163. Are these questions and analysis limited to year 2 students? If so, state it clearly.
Page8, lines 192-193. How many questions were there? Which ones measured attitude? Were they yes/no answers? Or by the Likert Scale?
Page 8, line 206. Was the overall vaccination attitude (66.8%) positive? If so, then state it.
Page 8, lines 207-208. Are you comparing to another study in the next sentence? Clarify this.
Page 8, lines 209-211. Again, you are comparing your study with the literature? Please state this more clearly in the whole paragraph.
Page 8, lines 213-217. This paragraph describes your study and the literature. Much better.
Page 9, lines 222-227. Did you compare two surveys taken by the same cohort of students? I may have missed something important here, how do you know about these increases?
Page 10, lines 253-255. In the limitations, it says you do not know who participated in the vaccination campaign of the second year students, but that some did. So I am not sure you can say there is a direct involvement unless you can specifically show those who participated did show an positive increase in benefits of vaccination. Please show this work in results if you are to say this.
Page 9, lines 260-261. I think you need to say this differently. ‘our goal is to increase vaccination and vaccine knowledge, which can be helpful through participating in a vaccination campaign’.
Page 10. References 6, 7, 8, and 26 need to be reformatted. They have been placed in by cut and paste.
Content questions
- Is the content relevant? Yes.
- Is the information accurate and complete? Please note if there is important information that is missing.
There are several areas that need clarification.
- Does the information flow in a logical fashion? If not, please suggest an alternative organization plan.
The manuscript flowed appropriately.
- Are there any key other references that should be added?
yes
- Do illustrations, figures, tables, etc. add to the content? Are they clear? Are there others you would recommend?
The tables and figures were appropriate, but more clarification on which group was depicted and why not the other group?
- Does the author provide implications?
See highlights.
General comments:
Thank you for the opportunity to review this article. The basis of this study is appropriate and warranted. Beyond the syntax and English language issues (yellow highlights), this was informative and support current literature even though it is not generalizable due to being conducted in one setting.
There are issues with clarifying the design, questionnaire, and the cross sectional comparison of both groups of study participants. See my comments throughout the manuscript. Much of it needs better explanation (green highlights).
The references support the manuscript.
Author Response
- Page 1, line 16; Page 1, line 18; Page 1, line 19; Page 1, line 21; Page 1, line 24 and 25; Page 1, line 33; Page 1, line 36; Page 1, line 39; Page 1, line 40; Page 1, line 41; Page 2, lines 43-44; Page 2, line 45; Page 2, line 52; Page 2, lines 53-54;
Following your suggestions, we made these corrections in the text.
- After the above Page 2, lines 53-54 and through the rest of this document, all the yellow highlights in the manuscript are syntax errors and need to be rewritten for clarification or for grammar editing. My questions will be in Green highlights, that are not necessarily considered syntax errors.
According to your suggestions, we improved the text.
- Page 3. Line 83.
We adjusted the text of the manuscript.
- Page 3, lines 107-110. This information on the difference between students in first year vs second year needs to be stated in section 2.3 Design and data collection. …..
The information you refer is now reported in section 2.1 “Nurses and midwives in the hospital vaccination team” and in section 2.2 “Design and data collection”.
- Page 3, line 124. This says the course. Is it one course or the program or year of study?
In Italy, the master’s degree in Nursing and Midwifery consists of 2-year course for professional nurses and midwives. We provided a specification in section 2.1 (row 71-73).
- Page 4, Table 1. Is there a difference between Degree and Postgraduation titles? Please explain in the narrative text.
According to your suggestion, we modified “post-graduation titles” in “post-graduate courses” in table 1, referring to the postgraduate education which involves learning and studying for academic or professional degrees, for which a first or bachelor's degree generally is required.
- Page 5, line 141. Please tell us which questions were evaluated in this assessment. One question? Several, please be specific.
Some questions from section A (1-5, 8, 14, 15) have been filed together to analyze the general attitude of nurses and midwives to flu vaccination. We explained the methodology we used for the evaluation of the vaccination attitude in row 105-107.
- Page 5, Table 2. Does this reflect the answers of all the students? Would it make sense to have two tables, one for the 1st year students and one for the 2nd year students? ….
The results showed in table 2 analyse the whole sample of nurses and midwives. This analysis was made to assess the general vaccination coverage of the sample. We decided to show only one comprehensive table, because there were no significant differences between the answers in section A, so we did not want to be redundant.
- Page 6, lines 147-150. I would like to see the questions that ‘vaccination attitude’ is derived from. Are they the numbers and percentages in Table 3 and/or Figure 2? Please tell us more as to how you ascertained this information.
Some questions from section A (1-5, 8, 14, 15) have been filed together to analyze the general attitude of nurses and midwives to flu vaccination. We explained the methodology we used for the evaluation of the vaccination attitude in row 105-107.
- Page 6, lines 154-163. Are these questions and analysis limited to year 2 students? If so, state it clearly.
These results are not limited to the second-year students but the whole sample of nurses and midwives was considered. This analysis was made to assess the general vaccination coverage of the sample. However, we added a specification in the sentence (row 158).
- Page 8, lines 192-193. How many questions were there? Which ones measured attitude? Were they yes/no answers? Or by the Likert Scale?
Some questions from section A (1-5, 8, 14, 15) have been filed together to analyze the general attitude of nurses and midwives to flu vaccination. We explained the methodology we used for the evaluation of the vaccination attitude in row 105-107.
- Page 8, line 206; Page 8, lines 207-208; Page 8, lines 209-211.
Following your suggestions, we improved the text of the manuscript.
- Page 9, lines 222-227 Did you compare two surveys taken by the same cohort of students? I may have missed something important here, how do you know about these increases?
Our results referred to the comparisons made between first- and second-year students, in terms of attitude and intention to get vaccination, and vaccination rates. However, we have clarified this further in the text (row 74-80, 84-87).
- Page 10, lines 253-255 In the limitations, it says you do not know who participated in the vaccination campaign of the second-year students, but that some did. So I am not sure you can say there is a direct involvement unless you can specifically show those who participated did show an positive increase in benefits of vaccination. Please show this work in results if you are to say this.
In order to assess the effectiveness of the new implemented measures on the students’ attitude to vaccination, the results were evaluated by comparing the answers of the second-year students (who did totally participate in the last vaccination campaign) with those of the first year (who did not participate).
Following your suggestion, we improved the limitations section, explaining our study limit related to the comparisons made between first- and second-year students (row 246-248).
- Page 9, lines 260-261 I think you need to say this differently. ‘our goal is to increase vaccination and vaccine knowledge, which can be helpful through participating in a vaccination campaign’.
We revised and improved the conclusions of the paper.
- Page 10. References 6, 7, 8, and 26 need to be reformatted. They have been placed in by cut and paste.
Following your suggestion, we revised the entire bibliography following the author guidelines of the journal.
- Content questions
- Is the content relevant? Yes.
- Is the information accurate and complete? There are several areas that need clarification.
- Does the information flow in a logical fashion? The manuscript flowed appropriately.
- Are there any key other references that should be added? Yes
- Do illustrations, figures, tables, etc. add to the content? The tables and figures were appropriate, but more clarification on which group was depicted and why not the other group?
- Does the author provide implications? See highlights
Thank you for reviewing the article, your revisions are accurate and helped us to improve the quality of the manuscript.
Reviewer 3 Report
This paper addresses an important aspect of health care. Vaccine hesitancy is a growing threat to disease prevention and control so assessing this amongst health care workers is critical.
Here are a few comments to improve on the article:
- Extensive English language improvements are required.
- In lines 111 to 112, the authors refer to statistical tests performed and indicate that Fisher's and chi-square tests were used to evaluating qualitative variables. I think these statistical tests are used for categorical variables. Qualitative variables have a different analysis approach and this is not a qualitative study.
- In lines 129 to 133 of the results section, the authors use the word "believed" a number of times when referring to the participants' responses to the questionnaire. The questionnaire did not ask about beliefs so I think it is better to use the expression "the participants reported" throughout the results section.
- In the results section, under 3.5-Opinions on the hospital vaccination campaign, the authors report mean ratings and standard deviations. Given that this score ranged from zero to ten, it is unlikely that this variable was normally distributed. If the authors want to report mean and standard deviation, they should demonstrate their statistical test for normality. Otherwise, they should report medians and ranges. This comment also applies to lines 199 to 202 where responses to question 9 of section C is reported.
- The questionnaire did not collect data on vaccine side-effects. I think this is an important factor to consider when addressing issues of vaccine hesitancy and should be addressed as a limitation to the study.
- The discussion can be improved by putting together the ideas in a more coherent manner.
- In the conclusion, the authors refer to the effects of the vaccination campaign on the study outcomes. This introduces some confusion because it seems to suggest that the study is evaluating an intervention (the campaign). If this is the case, then the study design would need some modifications. Otherwise, the authors should not refer to the campaign as an intervention that was evaluated using the questionnaire.
Author Response
- In lines 111 to 112, the authors refer to statistical tests performed and indicate that Fisher's and chi-square tests were used to evaluating qualitative variables. I think these statistical tests are used for categorical variables. Qualitative variables have a different analysis approach and this is not a qualitative study.
Following your suggestion, we reconsidered the variables you cite as categorical variables (row 112).
- In lines 129 to 133 of the results section, the authors use the word "believed" a number of times when referring to the participants' responses to the questionnaire. The questionnaire did not ask about beliefs so I think it is better to use the expression "the participants reported" throughout the results section.
According to your indications, we made changes you suggest (row 132-134).
- In the results section, under 3.5-Opinions on the hospital vaccination campaign, the authors report mean ratings and standard deviations. Given that this score ranged from zero to ten, it is unlikely that this variable was normally distributed. If the authors want to report mean and standard deviation, they should demonstrate their statistical test for normality. Otherwise, they should report medians and ranges. This comment also applies to lines 199 to 202 where responses to question 9 of section C is reported.
Following your suggestion, question 8 and 9 were analysed reporting medians and ranges, instead of mean and standard deviation. We also retested the relationship between questions 8-9 and the other variables by using the U of Mann-Whitney non-parametric test (row 112-114). Accordingly, section 3.5 and 3.6 have been fixed (row 170-184 and 197-204).
- The questionnaire did not collect data on vaccine side-effects. I think this is an important factor to consider when addressing issues of vaccine hesitancy and should be addressed as a limitation to the study.
The aim of the study was not to measure the effectiveness of the vaccination, neither to specify which are its side effects. The validated questionnaire was built and validated only to evaluate beliefs and knowledge about influenza vaccination, also including opinions referred to the side effects of the vaccination (some questions from section A and B).
- The discussion can be improved by putting together the ideas in a more coherent manner.
According to your suggestion, we made some improvements to the discussion section.
- In the conclusion, the authors refer to the effects of the vaccination campaign on the study outcomes. This introduces some confusion because it seems to suggest that the study is evaluating an intervention (the campaign). If this is the case, then the study design would need some modifications. Otherwise, the authors should not refer to the campaign as an intervention that was evaluated using the questionnaire.
The study shows how the involvement of nursing and midwifery students in the planning and execution of the vaccination campaign could influence their opinions and beliefs on flu vaccination among HCWs. The conclusion section was improved according to your suggestion.
Round 2
Reviewer 2 Report
Authors.
I appreciate the extensive changes you made to the manuscript.
I still have found more syntax errors. See below. Otherwise the manuscript is complete
Review #2
Page 1, line 19. Putting The in front of 86% still is not appropriate. Revise entire sentence. Page 1, lines 39-41. Revise again. In Italy, .... was the same as the previous years at 15.6%; Europe, Belgium ... were closer ... respectively. Page 2, line 43. ... 90.5% USA nurses receive vaccinations. Page 2 line 47. Knowledge is not plural. No s at end. Page 2, line 65. And to evaluate the effectiveness of the student involvement from the second year course. Page 3, line 125-126. Revise: there were 31 first year students and 30 second year students. Page 4, line 137. Instead of “The students whose reported that... “. revise to: students who reported their colleagues... Page 8, line 213. Knowledge is not plural. Page 8, line 221-222. ...”the two years students”. Should read between the first year and second year students”. Page 9, lines 224-225. Moreover... not a complete sentence. Page 9. Limitations. All sentences referring to “ limit” change each one to limitations.Please make above changes.
Author Response
Many thanks for your precious revision. We carefully analysed the syntax errors you reported and improved the manuscript.
Reviewer 3 Report
Thank you for the great work and for addressing the peer-review comments.
There is some amount of language editing required to improve on the quality of the paper.
Author Response
Many thanks for your precious revision. We carefully made an extensive language editing and improved the manuscript.